# The Oral Microbiota, Microbial Metabolites, and Immuno-Inflammatory Mechanisms in Cardiovascular Disease

**DOI:** 10.3390/ijms252212337

**Published:** 2024-11-17

**Authors:** Zheng Wang, Robert C. Kaplan, Robert D. Burk, Qibin Qi

**Affiliations:** 1Department of Epidemiology and Population Health, Albert Einstein College of Medicine, Bronx, NY 10461, USA; 2Public Health Sciences Division, Fred Hutchinson Cancer Research Center, Seattle, WA 98109, USA; 3Department of Obstetrics & Gynecology and Women’s Health, Albert Einstein College of Medicine, Bronx, NY 10461, USA; 4Department of Microbiology & Immunology, Albert Einstein College of Medicine, Bronx, NY 10461, USA; 5Department of Pediatrics, Albert Einstein College of Medicine, Bronx, NY 10461, USA; 6Department of Nutrition, Harvard T.H. Chan School of Public Health, Boston, MA 02115, USA

**Keywords:** oral microbiome, gut microbiome, cardiovascular diseases, inflammatory markers, microbial metabolites

## Abstract

Cardiovascular diseases (CVDs) remain a leading cause of global morbidity and mortality. Recent advancements in high-throughput omics techniques have enhanced our understanding of the human microbiome’s role in the development of CVDs. Although the relationship between the gut microbiome and CVDs has attracted considerable research attention and has been rapidly evolving in recent years, the role of the oral microbiome remains less understood, with most prior studies focusing on periodontitis-related pathogens. In this review, we summarized previously reported associations between the oral microbiome and CVD, highlighting known CVD-associated taxa such as *Porphyromonas gingivalis*, *Fusobacterium nucleatum*, and *Aggregatibacter actinomycetemcomitans*. We also discussed the interactions between the oral and gut microbes. The potential mechanisms by which the oral microbiota can influence CVD development include oral and systemic inflammation, immune responses, cytokine release, translocation of oral bacteria into the bloodstream, and the impact of microbial-related products such as microbial metabolites (e.g., short-chain fatty acids [SCFAs], trimethylamine oxide [TMAO], hydrogen sulfide [H_2_S], nitric oxide [NO]) and specific toxins (e.g., lipopolysaccharide [LPS], leukotoxin [LtxA]). The processes driven by these mechanisms may contribute to atherosclerosis, endothelial dysfunction, and other cardiovascular pathologies. Integrated multi-omics methodologies, along with large-scale longitudinal population studies and intervention studies, will facilitate a deeper understanding of the metabolic and functional roles of the oral microbiome in cardiovascular health. This fundamental knowledge will support the development of targeted interventions and effective therapies to prevent or reduce the progression from cardiovascular risk to clinical CVD events.

## 1. Introduction

Cardiovascular diseases (CVDs) remain a primary cause of mortality worldwide, responsible for an estimated 17.9 million deaths each year (data from WHO), and are also a leading cause of morbidity and disability [1]. The global prevalence of CVDs nearly doubled from 1990 to 2019, reaching approximately 523 million cases. During this same period, global trends for the burden of CVD-associated events, along with the years of life lost due to CVDs, showed a marked increase [1].

Over the past decade, advances in sequencing and multi-omics technologies (such as metabolomics and proteomics) have significantly enhanced our understanding of human microbiome–host interactions, including the potential role of the human microbiome in the development of CVDs [2,3]. The microbiome plays a crucial role in human metabolic function and is essential for the proper development of the immune system [4,5]. Microbiome dysbiosis, including the alterations of microbiome composition, distribution, microbial related function, or metabolic activities [6], have been linked to several CVD-related conditions, such as atherosclerosis, heart failure, hypertension, and type 2 diabetes [7,8].

The oral microbiota is the second-largest microbial community in the human body, surpassed only by the gut microbiota. More than 772 microbial species have been identified in the oral cavity [9]. The oral cavity contains several ecological niches, including saliva, the tongue, buccal mucosa, palate, dental surfaces, gingiva, and both subgingival and supragingival sites. Each of these niches hosts distinct microbial species with varying activities, and their susceptibility to disease is also different. The oral microbiota is involved in the preliminary digestion of food and produces a variety of primary and secondary metabolites [10,11]. The circulatory system provides a route for oral bacteria and their byproducts—including microbial metabolites and endotoxins—to enter the bloodstream and trigger systemic inflammation. In addition, these microbial components stimulate the release of host-derived pro-inflammatory cytokines as part of the immune response, intensifying inflammation and impacting various organs and tissues throughout the body [12]. Increasing evidence from epidemiological studies, clinical investigations, and basic science studies supports the interactions between oral bacteria, oral microbial products, and the development of cardiovascular inflammation and CVDs [13,14,15].

This manuscript reviews the existing evidence on the relationship between the oral microbiome and the progression of primary CVDs. We discuss the potential mechanisms by which the oral microbiome can affect cardiometabolic health, especially through oral microbiota-induced inflammation and immune responses, and the role of oral microbial metabolites. We then discuss future directions and oral microbiome-related strategies for CVD prevention and management.

## 2. Oral Microbiome and CVDs

### 2.1. The Interactions Among Periodontitis, Oral Microbiota, and CVDs

Previous studies have highlighted the role of specific oral bacteria in periodontal infections, and both epidemiological and mechanistic evidence have established links between periodontitis and CVDs (especially atherosclerosis and thromboembolic events) [16].

The bacterial biofilms on teeth and gums, commonly known as dental plaque, was known as primary etiological factors for the development of periodontitis, a chronic inflammatory condition characterized by the progressive destruction of gingival connective tissue, periodontal ligaments, and alveolar bone. These biofilms are predominantly composed of bacteria from the viridans group streptococci [17]. Additionally, several anaerobic bacteria such as *Porphyromonas gingivalis*, *Treponema denticola*, and *Tannerella forsythia*, play significant roles in chronic periodontitis [18,19]. Periodontitis serves as a reservoir for a wide range of microorganisms, which can enter the bloodstream through ulcerated, inflamed crevices and the pocket epithelium, allowing access to the adjacent gingival microcirculation [17]. The severity of bacteremia in patients with chronic periodontitis is directly related to the level of gingival inflammation [20]. The levels of *Streptococcus mutans* and *P. gingivalis* have been observed to be elevated in conjunction with the development of periodontal disease and systemic inflammation [21,22].

Meta-analyses of large population studies have identified periodontitis as a risk factor for both peripheral and carotid atherosclerotic CVD [23,24,25]. Studies have demonstrated that periodontitis significantly increases the risk of atherosclerotic plaque formation. A prospective study of the US population found that individuals with periodontitis had a 25% increased risk of developing atherosclerotic plaques [26]. Furthermore, data from the Atherosclerosis Risk in Communities (ARIC) study revealed that periodontal disease is significantly associated with an increased incidence of stroke, with a hazard ratio (HR) of 2.6 for cardioembolic stroke and 2.2 for thrombotic stroke [27]. Periodontitis was also linked to heart failure, coronary heart disease, and hypertension [28,29]. A recent study suggested that periodontitis may share overlapping metabolic pathways with atherosclerotic cardiovascular diseases, highlighting potential interconnected mechanisms [30].

Previous evidence has summarized the role of oral microbiota in the context of periodontitis. While the interactions between specific oral pathogenic bacteria and periodontitis, as well as the association between periodontitis and CVD, were well-documented, the direct relationship between the overall oral microbiota composition and CVD remains under-investigated. More research is needed to elucidate the precise mechanisms connecting oral microbiota to cardiovascular disease outcomes [31]. Further studies are also warranted to determine whether periodontal treatment holds the potential to reduce CVD risk and to reveal whether mechanisms related to the oral microbiota contribute to this possible protective effect.

### 2.2. Specific Oral Bacterial Taxa and CVD

The relative abundance of oral *Porphyromonas*, *Fusobacterium*, *Aggregatibacter*, and *Campylobacter* has been found to be associated with CVD in several human studies (Table 1). Many of these pathogenic bacteria share common characteristics: they are anaerobic or facultative anaerobic, have outer membrane vesicles, possess extracellular proteolytic activity, engage in the amino acids anaerobic fermentation, and produce harmful metabolites. These properties enable the bacteria to degrade periodontal tissue and infiltrate the bloodstream. Their tissue-destructive capabilities may explain the presence of these bacteria or the detection of bacterial DNA in serum and arterial walls. And many of these bacteria are also associated with periodontitis.

Bacterial material, such as DNA, has been identified in multiple cardiovascular structures, particularly within atherosclerotic plaques. Based on data from 1791 individuals across 63 studies, researchers identified 23 unique oral commensal bacteria present within the atherosclerotic plaques in individuals undergoing interventional procedures [32]. The in vivo identification of bacterial material within atherosclerotic plaques has drawn considerable interest. More compelling evidence comes from the cultivation of periodontal pathogenic bacteria from atheromatous plaques. Kozarov et al. found the presence of vital *P. gingivalis* and *Aggregatibacter actinomycetemcomitans* inside atheromatous tissue cultured with primary human coronary endothelial cells [33]. Furthermore, the in vitro ability of periodonto-pathogens to invade cardiovascular cells (including aortic and heart endothelial cells) has been established [34].

For the aforementioned oral bacteria, their ability to live in anaerobic tissue and their relationship with inflammation are considered important. However, their role in the development of CVD and the possible biological mechanisms are not fully understood.

**Table 1 ijms-25-12337-t001:** Human studies examining the association between cardiovascular diseases and the oral microbiome.

Author, Year	Location	Sample Size	Sample Site	Methods	Findings	Specific Quality Features
Hernández-Ruiz et al., 2024 [35]	Mexico	Patients with myocardial infarction (n = 16)	Supragingival dental plaque	16S, V3–V4	A positive and significant correlation of blood TMAO levels with oral *Porphyromonas* was identified in patients with myocardial infarction.	N
Rao et al., 2023 [36]	India	Patients with CAD (n = 12)	Subgingival plaque	16S, V3–V4	Twenty-two bacterial genera were shared between subgingival and atherosclerotic plaques, with *Acinetobacter* being dominant.	N
Schulz et al., 2021 [37]	Germany	Patients with CVD (n = 102)	Subgingival plaque	16S, V3–V4	One biomarker of *Saccharibacteria* phylum (class: TM7-3; order: CW040; family: F16) was associated with the incidence of a secondary CV event.	N
Leskelä et al., 2020 [38]	Germany	Controls (n = 100), patients with stroke (n = 98); total (n = 198)	Saliva	Targeted qPCR sequencing	Specific oral bacteria *A. actinomycetemcomitans* concentration ↑ in ischemic stroke cases in comparison to controls. IgG against *A. actinomycetemcomitans* is one of the main determinants of LPS neutralizing capacity.	Cases matched controls in terms of age and sex.
Perry et al., 2020 [39]	New Zealand	Patients with atherosclerosis (n = 100)	Saliva	Targeted qPCR	Acute stroke patients were at increased risk of colonization from respiratory pathogens. The presence of these pathogens in saliva in one month was associated with adverse respiratory events.	N
Nikolaeva et al., 2019 [40]	Russia	Patients with angina pectoris (n = 15), with acute myocardial infarction (n = 15), with chest pain but no CAD (n = 15); total (n = 45)	Oral plaque	Targeted 16S sequencing	In acute myocardial infarction patients, the frequency of *P. gingivalis*, *T. forsythia*, and *A. actinomycetemcomitans* detection was significantly higher than in participants without CVD.	N
Su et al., 2019 [41]	Japan	Total sample (n = 70)	Tongue dorsum	Targeted PCR sequencing	Subjects with medical histories of stroke and heart disease exhibited a trend toward higher *P. gingivalis* positive rates on the tongue dorsum than those without such disorders.	N
Liljestrand et al., 2018 [42]	Finland	Controls (n = 123), stable CAD (n = 184), ACS (n = 169), ACS-like no CAD (n = 29); total (n = 505)	Subgingival plaque	DNA hybridization	Periodontal pathogens such as *A. actinomycetemcomitans* and the antibody levels to these pathogens associated with coronary artery disease and acute coronary syndrome.	Multivariable adjustment for age, gender, and CVD risk factors.
Ziebolz, Rost et al., 2018 [43]	Germany	Patients undergoing surgery for aortic valve stenosis (n = 10)	Subgingival plaque	Targeted PCR sequencing	Demonstrate the presence of periodontal bacteria DNA in human cardiac tissue. Identified correlations of inflammatory proteins and infection markers with valvular heart disease.	N
Ziebolz, Jahn et al., 2018 [44]	Germany	Patients undergoing surgery for aortic valve stenosis (n = 30)	Subgingival plaque	Targeted PCR sequencing	Periodontal pathogen (e.g., *P. gingivalis*, *C. rectus*, *P. intermedia*, *F. nucleatum*) DNA found in atrial and myocardial tissue and linked to tissue inflammation.	N
Kannosh et al., 2018 [45]	Serbia	Patients with atherosclerosis (n = 100)	Subgingival plaque	Targeted 16S sequencing	Detect presence of periopathogenic bacteria in subgingival and atherosclerotic plaque (*P.gingivalis*, *P.intermedia*, *T. forsythensis*). Patients’ ages could have influenced the findings.	N
Mahalakshmi et al., 2017 [46]	India	Patients with atherosclerosis (n = 65), with periodontitis but no systemic disease (n = 59), controls (n = 100); total (n = 224)	Subgingival plaque	Targeted 16S sequencing	Statistical significance was observed for the prevalence of 16S rRNA of *P. gingivalis*, *T. forsythia*, *T. denticola*, and *P. nigrescens* both in subgingival plaque and atheromatous plaque. Significant odds and risk ratio to atherosclerosis were observed for these bacteria.	N
Boaden et al., 2017 [47]	UK	Patients with stroke (n = 50)	Saliva, buccal mucosa, tongue, gingiva, hard palate	16S, V1–V9	Described the bacterial profile of the oral flora during the first 2 weeks following a stroke; 14 of the 20 most common bacterial phylotypes found in the oral cavity were *Streptococcal* species, with *S. salivarius* being the most common. The condition of the oral cavity worsened during the study period.	N
Mougeot et al., 2017 [48]	USA	Patients with CVD (n = 42)	Coronary artery tissue and femoral artery tissue	16S, V3–V4	The most abundant species were *P. gingivalis*, *E. faecalis*, and *F. magna*.	N
Fåk et al., 2015 [49]	Sweden	Asymptomatic atherosclerosis (n = 35), symptomatic atherosclerosis (n = 27), controls (n = 30); total (n = 92)	Whole mouth swab	16S, V1–V2	Abundance of *Anaeroglobus* in the oral cavity could be associated with symptomatic atherosclerosis.	Cases matched controls in terms of age and sex.
Serra e Silva Filho et al., 2014 [50]	Brazil	Patients with periodontitis and atherosclerosis (n = 18)	Subgingival plaque	16S, V1–V9	Periodontal pockets and atheromatous plaques of cardiovascular disease patients can present similarities in the microbial diversity. Seventeen identical phylotypes (including *P. gingivalis*, *T. vincentii*, *F. nucleatum*) were found in atheroma and subgingival samples, indicating possible bacterial translocation.	N
Koren et al., 2011 [51]	Sweden	Controls (n = 15), patients with atherosclerosis (n = 15); total (n = 30)	Oral cavity swab	16S, V1–V2	16S rRNA sequencing identified *Chryseomonas*, *Veillonella*, and *Streptococcus* in atherosclerotic plaque samples. The combined abundances of *Veillonella* and *Streptococcus* in atherosclerotic plaques correlated with their abundance in the oral cavity.	The control subjects were matched to the patient group by sex.

N, not applicable. ↑, increased.

#### 2.2.1. *Porphyromonas gingivalis*

*P. gingivalis*, a Gram-negative oral anaerobe, is widely recognized as a major etiologic agent responsible for the onset and progression of severe periodontitis [52]. *P. gingivalis* has been detected in both subgingival samples and coronary artery atherosclerotic plaques [46,50]. Elevated levels of *P. gingivalis* in oral samples have been observed in patients with coronary heart disease [40,45,46,50], coronary artery disease [49], valvular heart disease [43], and in individuals with CVD history [41]. Evidence also suggested that *P. gingivalis* is linked to the citrullination of host self-antigens, contributing to the pathogenesis of rheumatoid heart disease [17]. Animal studies have indicated that *P. gingivalis* can accelerate atheroma plaque formation and has the ability to induce the development of fatty streaks in the aorta of rabbits [53].

*P. gingivalis* is also known for its ability in disrupting innate immune function and triggering inflammatory responses [52]. The lipopolysaccharide of *P. gingivalis* can specifically activate host immune defenses, leading to heightened inflammatory pathways. *P. gingivalis* exhibits convertase-like enzymatic activity and manipulates complement-TLR interactions to subvert host immune defenses, enabling it to evade immune clearance. This allows the pathogen to persist within the host, contributing to a shift in the periodontal microbiota toward a dysbiotic state, ultimately leading to inflammatory periodontitis [52]. Studies based on serum concentrations of anti-*Porphyromonas* antibodies have shown that several *Porphyromonas* species are positively associated with cardiovascular risk [54,55].

#### 2.2.2. *Fusobacterium nucleatum*

*Fusobacterium* is a Gram-negative anaerobic genus commonly residing in both oral cavity and gastrointestinal tract [51,56]. In the oral cavity, it is particularly abundant in dental plaque [57]. Elevated levels of *Fusobacterium* have been reported in subgingival samples of patients with valvular heart disease [43,44]. Additionally, multiple species of *Fusobacterium* have been detected in subgingival plaque, as well as in coronary artery atherosclerotic plaques [50], and carotid artery plaque tissues [32,51,58]. Both gut and oral Fusobacterium might be potential sources of Fusobacterium in carotid artery plaque [59,60]. As the most representative and thoroughly studied species in this genus, F. nucleatum has been associated with an increased risk of carotid artery atherosclerotic plaque [59,60]. Recent research indicated that antibodies to *F. nucleatum* may be linked to cardiovascular events. A study found that higher antibody level against *F. nucleatum* was significantly associated with higher incidence of coronary heart disease, after adjustment for established risk factors [61]. Recent integrated omics studies using gut microbiome and serum proteomics data indicated that. *F. nucleatum* was positively associated with several serum proteomic markers which might be involved in host inflammation and immune activation related to bacterial infection (e.g., CXCL9, TNFRSF9), and the association between *F. nucleatum* and carotid artery plaque can be partially explained by these serum inflammatory markers [59,60]. Furthermore, Fusobacterium is proteolytic and forms cytotoxic end products. It has been associated with multiple plasma metabolites, such as lysophosphatidylcholines (LPCs), lysophosphatidylethanolamines (LPEs), and diacylglycerols (DGs), which have been further linked to an increased risk of carotid artery plaque [59].

#### 2.2.3. *Aggregatibacter actinomycetemcomitans*

*A. actinomycetemcomitans* is a facultative anaerobic, Gram-negative rod that takes on a coccoid shape. The facultative anaerobic feature enables it to act as an early colonizer of the periodontium. It is frequently detected in the oral cavity, particularly in subgingival plaque from individuals with aggressive periodontitis [57]. This bacterium has also been identified in samples from both subgingival plaque and atherosclerotic lesions in coronary arteries [45], indicating its potential involvement in cardiovascular pathology.

The relative abundance of *A. actinomycetemcomitans* has been reported to increase in saliva samples from patients with ischemic stroke [38]. Similarly, higher levels of *A. actinomycetemcomitans* have been detected in subgingival or periodontal pocket samples from individuals with acute coronary syndrome, coronary artery disease [42], and valvular heart disease [43,44]. One study found that Elevated serum concentrations of anti-*A. actinomycetemcomitans* antibodies were linked to a greater cardiovascular risk and future coronary heart disease events [55].

The subspecies of *A. actinomycetemcomitans* exhibit varying characteristics, such as the ability to induce leukocyte and monocyte cytotoxicity. For example, *A. actinomycetemcomitans* hold the ability to produces leukotoxin (LtxA), a member of the RTX (repeats-in-toxin) family, which specifically targets human leukocytes by binding to the β2 integrin lymphocyte function-associated antigen-1 (LFA-1) on white blood cells, leading to cell death [62]. This interaction undermines host immune defenses, facilitating immune evasion and enhancing the bacterium’s pathogenic potential. Additionally, these bacteria can produce immunoglobulin proteases and collagenase, which contribute to their pathogenic potential and tissue destruction [57].

#### 2.2.4. *Prevotella intermedia*

*P. intermedia* is a slow-growing, anaerobic, Gram-negative rod that exhibits a coccoid shape [57]. *P. intermedia*, along with other *Prevotella* species such as *Prevotella nigrescens* and *Prevotella loescheii*, have been detected in both oral samples (e.g., subgingival plaque) and coronary artery atherosclerotic lesions [63]. Other studies have reported positive associations between various *Prevotella* species (both in whole mouth samples [49] and subgingival samples [46,50]) and coronary artery disease. Additionally, *Prevotella* species have been associated with valvular heart disease [43,44], highlighting the broader role of this genus in cardiovascular pathology. Among the various *Prevotella* species, *P. intermedia* is one of the most extensively studied in relation to CVDs. *P. intermedia* produces lipase during the bacterial invasion of the periodontium and exhibits a proteolytic metabolism, breaking down proteins to support its growth and virulence within host tissues [57]. This enzymatic activity plays a key role in contributing to inflammatory processes and tissue degradation.

#### 2.2.5. *Treponema denticola*

*T. denticola* is an oral anaerobic bacterial species associated with chronic periodontitis [64]. The main site for *T. denticola* habitation in the oral cavity is the gingival crevice. In patients with coronary artery disease, the relative abundance of *Treponema* species were found to be increased in subgingival plaque and whole mouth samples [49]. It is of note that *T. denticola* has been detected in both coronary artery plaque samples and subgingival samples [40,45,46], further suggesting the potential involvement of *T. denticola* in coronary artery disease.

*T. denticola* is known as an invasive spirochete. Previous animal studies indicated that it can penetrate gingival tissues and circulate through blood vessels, with the potential to invade the heart and cardiovascular endothelium in medium to large arteries, including the aorta, coronary, and carotid arteries [65]. A key virulence factor of *T. denticola* is the chymotrypsin-like proteinase (Td-CTLP), which activates matrix metalloproteinases, particularly pro-MMP-8 and pro-MMP-9, facilitating the degradation of extracellular matrix components. This enzymatic activity plays a pivotal role in inflammation and tissue remodeling. Moreover, Td-CTLP has been found to degrade important proteinase inhibitors, including TIMP-1, TIMP-2, and α-1-antichymotrypsin, as well as the complement component C1q, further contributing to immune evasion and tissue damage.

Several other oral microbial species, including *Tannerella forsythia*, *Campylobacter rectus*, *Parvimonas micra*, and *Eubacterium timidum*, have also been linked to oral dysbiosis and cardiovascular disease. However, most current research on the relationship between oral microbiome and CVDs has concentrated on bacteria taxa specifically associated with periodontitis [66,67]. This limited focus highlights the need for broader investigations into the role of the entire oral microbiome (e.g., at community level) in cardiovascular health. Further studies are also warranted to explore the role of other microbial taxa of the oral microbiome community in CVDs and to elucidate the potential biological mechanisms involved.

## 3. Oral Microbiome and Gut Microbiome

Similarities have been detected between the compositions of the oral and gut microbiota [68]. It is estimated that approximately 45% of the microbial species may overlap between these two ecosystems [69]. The oral microbiota primarily consists of five dominant phyla: *Proteobacteria*, *Firmicutes*, *Bacteroidetes*, *Actinobacteria*, and *Fusobacteriota* [70]. Some of these phyla, such as Firmicutes, Bacteroidetes, and Actinobacteria, are also predominant in the gut microbiota. The predominant genera in the oral cavity may vary across populations from different geographical locations. For example, *Neisseria* is the dominant genus in the Chinese population, *Veillonella* in the Canadian population, and Prevotella in the Qatari population [71,72].

Currently, the mechanisms of oral–gut microbiota interactions in the progression of CVDs remain not fully understood. Evidence suggests that oral microbiota can trigger gut dysbiosis either through direct translocation to the gut or via other systemic pathways. In a large-scale study analyzing oral and gut microbiota across participants from five countries, bioinformatic analyses demonstrated that approximately 10% of the oral microbiota successfully transfers and establishes itself in the gut [73]. Our previous research on the gut microbiome identified that the well-known oral bacteria, *F. nucleatum*, an opportunistic pathogen, can also be found in the gut. *F. nucleatum* in the gut has been positively associated with arterial plaque formation [59,60].

Most current studies are animal-based, typically involving the oral administration of microbiota to mice and then investigating subsequent changes in gut microbiota. For example, after introducing *F. nucleatum* orally to healthy mice, researchers observed changes in the fecal microbiota, including an increased abundance of *F. nucleatum* and elevated levels of autophagy markers in colorectal tissues. This effect was mitigated or eliminated by the administration of antibiotics, such as metronidazole [74].

The transfer of opportunistic pathogens has been observed to occur more frequently among individuals with underlying disease conditions. The presence of certain specific pathogens, such as *F. nucleatum* subspecies, may also promote this transfer and subsequently worsen the severity of the disease [74].

Other examples highlighting the link between the oral and gut microbiomes in relation to CVDs include the oral-mediated alterations in the proportions of gut *Bacteroidetes* and *Firmicutes*, which serve as important biomarkers in patients with coronary heart disease and stroke [75,76]. In animal studies, the oral administration of *P. gingivalis* has been shown to influence gut microbial composition, particularly altering the proportions of *Bacteroidetes* and *Firmicutes*. It was shown that introducing *P. gingivalis* leads to a clear separation of gut microbiota composition, with an elevation in *Bacteroidetes* and a reduction in *Firmicutes*, coupled with enhanced intestinal permeability and compromised barrier function [77]. In C57BL/6 mice, oral administration of *P. gingivalis* resulted in a decrease in the relative abundance of *Bacteroidetes* in gut [78].

Thus, we hypothesize that the oral microbiota may influence the gut microbiota and contribute to the progression of CVDs, with significant therapeutic potential. For example, oral microbiota transplantation (OMT) has shown promise in mitigating detrimental alterations in both oral and gut bacteria [67], suggesting a potentially positive impact on CVD management.

## 4. Potential Mechanisms by Which the Oral Microbiota Affect CVDs

### 4.1. Oral Microbial Translocation

One plausible mechanism by which oral microbiota dysbiosis affects the development of CVD is through the translocation of oral bacteria, particularly the invasion of pathogenic bacteria into the circulation [17]. Oral inflammation enhances the permeability of periodontal blood vessels, facilitating the entry of oral bacteria into the systemic circulation, where they colonize atherosclerotic plaques and intensify inflammatory responses.

Previous human and animal studies have provided substantial evidence supporting this mechanism. Traces of DNA, RNA, or antigens from oral commensal bacteria, including *P. gingivalis*, *A. actinomycetemcomitans*, and *Veillonella* species [45,51,79], have been identified in atheromatous plaques, suggesting that various pathogenic species are capable of reaching and colonizing these affected sites. Importantly, in 2005, Kozarov et al. found the presence of vital *P. gingivalis* and *Actinobacillus actinomycetemcomitans* inside atheromatous tissue cultured with primary human coronary endothelial cells, providing strong evidence for the oral bacteria translocation hypothesis [33]. A previous study has identified 23 unique oral commensal bacteria present within the atherosclerotic plaques, in individuals undergoing interventional procedures [32].

Evidence from animal models highlights the causal role of *P. gingivalis* invasion in arterial endothelial cells, contributing to atherosclerotic plaque formation. In *P. gingivalis*-inoculated mice, the administration of the antibiotic metronidazole completely prevented the development of atherosclerotic lesions in mice fed a normal diet, and notably diminished the severity and extent of lesions in those on a high-fat diet [80].

In addition, another possible mechanism is that the oral microbiota may contribute to gut dysbiosis through oral–gut microbiota transfer, which increases gut permeability and promotes bacterial translocation. These changes are associated with chronic inflammation [17]. This process allows endotoxins to breach compromised gut membranes and directly enter the circulation, ultimately contributing to the development of CVDs [81].

### 4.2. Inflammation and Immune Responses

Evidence has demonstrated that alterations in the oral microbiota significantly contribute to both local and systemic inflammation, impacting cardiometabolic health and accelerating the development of CVDs [82]. Oral microbiota dysbiosis and elevated levels of periodontal pathogens induce immune responses, including the activation of neutrophils and macrophages. These immune responses also engage cells such as dendritic and gamma delta cells, which subsequently release pro-inflammatory mediators [83]. The persistence of this inflammatory response is closely linked to oral microbiota imbalance, perpetuating a harmful feedback loop [83].

Oral inflammatory lesions and elevated levels of specific oral bacteria lead to the release of pro-inflammatory cytokines into the bloodstream, promoting atherosclerotic plaque formation through the propagation of these inflammatory mediators [17]. This systemic inflammatory state is featured by increased levels of acute-phase proteins, pro-inflammatory cytokines and chemokines such as interleukin (IL)-6, and fibrinogen [84]. Additionally, oral bacteria entering the bloodstream and disruptions to gut microbiota through oral–gut microbial transfer may intensify systemic inflammation and contribute to the progression of cardiovascular disease [73,77].

#### 4.2.1. Lipopolysaccharide

Lipopolysaccharide (LPS), commonly known as endotoxin, is a component of the outer membrane of Gram-negative bacteria. Previous studies have highlighted the significant role of LPS in promoting inflammation which in turn may contribute to the progression of atherosclerosis. LPS acts as an immunostimulator or immunomodulator that can translocate from the oral cavity into the circulation, causing endotoxaemia and endothelial dysfunction [85]. Once in the circulation, LPS undergoes a transfer cascade, first binding to lipopolysaccharide binding protein (LBP), then to CD14, and finally to toll-like receptor 4 (TLR4). This binding sequence activates an inflammatory response involving the release of pro-inflammatory cytokines from targeted cells [86]. TLR4 has been detected in human atherosclerotic plaques and has been shown to facilitate atherosclerosis development in mouse models [87,88]. Moreover, elevated serum levels of LPS or LBP have been linked to an increased risk of cardiovascular disease [89,90]. Previous reports indicated that the keystone periodontal pathogen *P. gingivalis* secretes LPS, which interferes with polymorphonuclear leukocyte function by interacting with adhesion molecules like IL-8, ICAM-1, and E-selectin. This interaction disrupts leukocyte recruitment and hampers the immune response [91,92].

#### 4.2.2. Cytokines

Pro-inflammatory cytokines are critical mediators that link the oral microbiome to inflammatory atherosclerosis. Epidemiological evidence has demonstrated that elevated levels of cytokines like IL-1, IL-6, and TNF are associated with increased cardiovascular risk [93,94]. Many of these pro-inflammatory cytokines are linked with known oral pathogens. For example, in human studies using 16S sequencing data, oral *Porphyromonas* was positively associated with salivary cytokines such as IL-1β, IL-2, IL-8, and IL-13, while *Fusobacterium* was associated with IL-1β [95]. Additionally, oral *Streptococcus* species showed positive correlations with multiple cytokines, including IL-1β, IL-2, IL-4, IL-6, IL-7, IL-9, IL-12, and IL-17 [96].

The well-studied oral pathogen *P. gingivalis* expresses autoinducer 2, which induces the secretion of IL-8 in oral epithelial cells [97]. This cytokine, along with IL-6 and MCP1, is linked to endothelial dysfunction, characterized by increased procoagulant properties, mononuclear cell adhesion, and the elevated expression of cell adhesion molecules [98,99]. Additionally, both *P. gingivalis* and *F. nucleatum* can act on macrophages, neutrophils, and monocytes to induce the production of TNF-α, IL-6, and IL-8 [96]. An animal study indicated that in mice, infection with *P. gingivalis* induces the accumulation of macrophages and inflammatory mediators like CD40, IFN-γ, IL-1β, IL-6, and TNF-α in atherosclerotic lesions, with these responses being milder in immunodeficient mice [100].

Elevated concentrations of bacterial surface molecules stimulate the production of various inflammatory mediators and cytokines, fueling both local and systemic inflammatory responses [101,102]. These processes are driven through the activation of several inflammatory pathways, including matrix metalloproteinase 9 (MMP9), Nuclear factor kappa-B (NF-κB), and basic helix–loop–helix ARNT-Like 1 (BMAL1) [22,103]. Under the influence of pro-inflammatory agents such as TNF-α, IL-6, and transforming growth factor β (TGFβ), epithelial and immune cells are prompted to produce reactive oxygen species (ROS), reactive nitrogen species, and matrix metalloproteinases, which in turn activate the NF-κB signaling pathway, further amplifying the inflammatory response [104].

In addition to the aforementioned specific pathogens, other oral bacteria are also capable of inciting harmful inflammation that engages both the innate and adaptive immune systems [105]. Current evidence underscores the interactions between the oral microbiota and pro-inflammatory cytokines. However, the specific underlying mechanisms—such as whether, or to what extent, leakage from the periodontium or bacterial translocation may contribute to host pro-inflammatory cytokine levels—remain to be thoroughly investigated.

### 4.3. Modulation of Platelet Aggregation

One potential mechanism by which the oral microbiota influences the progression of CVD is through modulation of platelet aggregation, which can exacerbate inflammatory responses and contribute to the development of atherosclerosis and thromboembolic events.

Oral bacteria can stimulate platelets either by direct interaction and activation or indirectly by releasing platelet-activating factors. For instance, viridans group streptococci (VGS), such as *Streptococcus sanguinis*, *Streptococcus gordonii*, *Streptococcus mutans*, and *Streptococcus mitis*, have been shown to promote platelet adhesion and aggregation in vitro via various surface proteins, including platelet aggregation-associated protein (PAAP), serine-rich glycoproteins, adhesins, and glucosyltransferases [106,107,108,109]. Moreover, the immune response triggered by oral bacteria can induce platelet activation [107]. This process contributes to localized thrombus formation, the depletion of platelets, and an increase in the secretion of pro-inflammatory cytokines and mediators by the activated platelets, thus playing a role in promoting inflammation, atherogenesis, and thrombosis [110].

### 4.4. Oral Microbial Metabolites

#### 4.4.1. Trimethylamine N-Oxide

Trimethylamine N-oxide (TMAO) is recognized for its detrimental role in promoting the progression of cardiovascular disease. TMAO affects cardiovascular health by inhibiting reverse cholesterol transport, reducing bile acid synthesis [111], promoting platelet reactivity and thrombosis potential [112,113], and instigating vascular inflammation [114] TMAO is generated through microbial and host biochemical reactions, starting with the production of trimethylamine (TMA) from dietary choline and carnitine. While the gut microbiome is the primary source of TMA, emerging evidence suggests that oral bacteria, especially *Prevotella* and *Fusobacterium*, can also produce TMA. Studies have shown that patients with periodontal disease have increased TMAO levels, linking oral pathogens to increased cardiovascular risk [115]. Additionally, a recent study highlighted the association between the oral microbiome and TMAO levels in patients with myocardial infarction [35].

#### 4.4.2. Short-Chain Fatty Acids

SCFAs are important metabolites produced by the microbiota, playing a vital role in the host’s regulation of gluconeogenesis, lipid metabolism, and inflammatory response [116]. Bacterial species in the oral cavity, such as *Streptococcus*, *Actinomyces*, *Lactobacillus*, *Propionibacterium*, and *Prevotella* [117], can utilize carbohydrate-active enzymes to degrade carbohydrates into SCFAs, which sustain their energy requirements [118]. Dietary patterns, particularly high sugar intake, significantly influence the oral microbiota composition and SCFA production levels [119]. There is conflicting evidence regarding the effects of SCFAs in the oral cavity and the circulatory system. On one hand, SCFAs have anti-inflammatory effects in the plasma by inhibiting the NF-kB and Akt signaling pathways and lowering cytokine levels [76]. Moreover, they inhibit histone deacetylases (HDACs) and engage specific G protein-coupled receptors (GPRs), offering cardiovascular protection [120]. On the other hand, SCFAs can also affect the expression of connexins and adhesion proteins, which may compromise oral epithelial cell function [121], while still exerting systemic anti-inflammatory benefits that contribute to cardiovascular health.

#### 4.4.3. Nitric Oxide

Nitric oxide (NO) can play a beneficial role in vascular function due to its vasodilatory effects. It helps lower blood pressure, protect endothelial cells, and counteract the progression of atherosclerosis. Additionally, NO contributes to inflammation regulation, providing cardiovascular benefits by reducing inflammatory responses in blood vessels [122]. A deficiency in NO has been strongly linked to the development of cardiovascular disease, making it a significant marker for predicting cardiovascular events [123]. The oral microbiota contributes substantially to the production of NO, serving as a vital reservoir for NO within both the bloodstream and tissues [124]. The insufficient synthesis of NO by endogenous nitric oxide synthase (NOS) can be compensated by nitrite (NO_2−_) produced by oral microbiota. Current evidence indicates that alterations in the oral microbiota can significantly impact NO levels, thereby affecting the progression of CVD [125].

#### 4.4.4. Hydrogen Sulfide

Hydrogen sulfide (H_2_S) is a crucial endogenous gaseous signaling molecule involved in cardiovascular function [126]. It exerts antioxidant and anti-inflammatory effects, improves insulin resistance, and helps regulate blood pressure and atherosclerosis [127]. Elevated H_2_S levels have been linked to protection against hypertension and diabetic cardiomyopathy [128]. It may interact with nitric oxide (NO) to provide additional cardiovascular benefits [129].

However, a paradox arises with H_2_S in the oral cavity. While systemic H_2_S provides protective effects, increased H_2_S levels in the oral environment have been linked to heightened oral inflammation [130]. The oral microbiota, particularly proteolytic bacteria such as Prevotella and Porphyromonas, have the ability to produce H_2_S from sulfur-containing amino acids [126]. In healthy individuals, H_2_S levels are low, but in disease states where microbial loads increase, H_2_S production is significantly elevated [131]. The impact of oral microbiota-derived H_2_S on systemic H_2_S levels and cardiovascular health requires further investigation [132].

### 4.5. The Potential Influences of Genetic, Environmental, and Other Factors on Oral Microbiota–CVD Interaction

Recent omics studies have revealed that host genetic variations were associated with the composition of the oral microbiome and may have the potential to affect the growth of specific oral bacteria. Polymorphisms in genes such as leucine zipper motif isoform 2 (APPL2) and glucose transporter 9 (SLC2A9), which are closely linked to obesity and insulin resistance, have been associated with changes in the abundance of various oral bacteria, including *Prevotella* species. These genetic variations may regulate bacterial growth through mechanisms involving host microRNAs [133,134]. On the other hand, alterations in microbiota composition and function have also been implicated in disease pathogenesis through epigenetic modifications, such as DNA methylation, histone changes, and regulation by noncoding RNAs [135].

Many external factors, including environmental exposures, health behaviors, and socioeconomic status, are all potential contributors to oral microbiome composition and may play roles in modulating the relationship between the oral microbiome and CVD risk. For example, several bacteria enriched in the oral microbiomes of smokers, such as *Porphyromonas*, *Prevotella*, *Treponema*, and *Veillonella*, have also been positively associated with cardiovascular outcomes [63,136]. Thus, it is important for future research to consider multiple external factors and integrate multidisciplinary approaches, to more effectively elucidate the complex mechanisms underlying the oral microbiota–CVD interaction.

## 5. Research Gaps and Future Directions

While substantial evidence links the human microbiome to multiple CVD outcomes, most research has focused on the gut microbiome, with less known about the relationship between the oral microbiome and CVDs, despite growing interest in this field. The oral environment differs significantly from the intestinal environment, so mechanisms linking the gut microbiome and its products to CVDs may not fully apply to the oral cavity. Additionally, the oral microbiota, positioned upstream of the digestive tract, can also influence gut microbiota composition and function. Distinct mechanisms may be involved in how the gut and oral microbiomes contribute to the progression of CVDs, which warrants further investigation.

Importantly, most current studies on the link between oral bacteria and CVDs are cross-sectional and often focus on specific bacteria, especially pathogens associated with periodontitis or other oral diseases. Two important aspects remain underexplored. First, how do alterations in the overall oral microbiome community, as well as oral bacteria other than periodontal pathogens, contribute to the development of CVDs? Second, what are the mechanisms by which oral dysbiosis or oral microbiome alterations, rather than periodontal disease, influence the development of CVDs? Further investigations, including large-scale prospective population studies and intervention studies, are needed to establish causal relationships between the oral microbiome and CVDs and determine the biological validity of the proposed pathophysiological mechanisms discussed in this review.

The integrative analysis of multi-omics techniques (e.g., metagenomics, metatranscriptomics, metabolomics, and proteomics) may improve the current understanding. Integrated multi-omics studies, along with standardized and comparable methodologies, will facilitate a deeper and comprehensive understanding of the metabolic and functional roles of the oral microbiome in cardiovascular health. This approach will help clarify some of the hypothesized inflammatory mechanisms connecting oral microbiome alterations to CVD development. Moreover, it will contribute to the development of targeted interventions and effective therapies aimed at preventing or reducing the progression from cardiovascular risk to clinical CVD events.

## 6. Conclusions

In summary, current evidence indicates the significant role of the oral microbiome in the development and progression of CVDs. This review highlighted several well-established CVD-associated oral bacteria (e.g., *P. gingivalis*, *F. nucleatum*, and *A. actinomycetemcomitans*), and discussed the mechanisms underlying these associations. The potential mechanisms by which the oral microbiota can influence CVD progression include oral and systemic inflammation; immune responses and cytokine release; oral bacteria translocation; and microbial-related products, such as metabolites (e.g., TMAO, SCFAs, NO, H_2_S etc.) and toxins (e.g., LPS). Despite substantial progress in understanding these mechanisms, considerable gaps in knowledge remain, particularly regarding the impact of overall oral microbiome alterations beyond periodontal pathogens. Future research should aim to unravel these complex interactions through large-scale, prospective studies and multi-omics approaches. These efforts will be essential for developing targeted interventions and effective therapies to mitigate the risk and progression of CVDs, leading to more effective strategies for CVD prevention and management.

## Data Availability

No new data were created or analyzed in this study.

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
