# Peer review of "The Oral Microbiota, Microbial Metabolites, and Immuno-Inflammatory Mechanisms in Cardiovascular Disease"

_ijms, 2024, doi:10.3390/ijms252212337_

Round 1

Reviewer 1 Report

Comments and Suggestions for Authors

Review for IJMS – “The oral microbiota, microbial metabolites, and immuno-inflammatory mechanisms in cardiovascular disease”

Thank you for the invitation to review this article. This is a detailed and informative review on the mechanisms involved in the link between oral microbiota, microbial metabolites, and immune-inflammatory pathways. Please find my review and minor suggestions below:

1-     The authors missed an opportunity to discuss genetic susceptibility and how it may impact the oral microbiome as well as susceptibility to CVD. I suggest the inclusion of a brief introduction to this important subject. 

2-     Similarly, external confounders that play a role in the link between oral microbiome and CVD where not mentioned. For example, smoking and social determinants of health.

3-     Please add a reference to line 211 to the end of the sentence: “Prevotella intermedia produces lipase during bacterial invasion…”

4-     There is some additional recent literature that could be considered about metabolomics, microbiome, oral disease, and cardiovascular disease intersection:

https://www.ahajournals.org/doi/full/10.1161/JAHA.123.033350

https://link.springer.com/article/10.1007/s00784-021-04083-4

https://journals.sagepub.com/doi/10.1177/23800844241280383?url_ver=Z39.88-2003&rfr_id=ori:rid:crossref.org&rfr_dat=cr_pub%20%200pubmed

5-     Please add a footnote to Table 1 to describe abbreviation of "N" in specific quality features.

Author Response

Thank you very much for taking the time to review this manuscript. Please find the detailed responses below and the corresponding revisions were highlighted in red font in the re-submitted files

Comments 1     The authors missed an opportunity to discuss genetic susceptibility and how it may impact the oral microbiome as well as susceptibility to CVD. I suggest the inclusion of a brief introduction to this important subject. 

Response 1: We greatly appreciate these constructive suggestions. As suggested, in this revision, we have added one specific section (i.e. section 4.5) to discuss the potential influences of genetic susceptibility on oral microbiota-CVD interaction. [Line 487-]
“Recent omics studies have revealed that host genetic variations were associated with the composition of the oral microbiome, and may hold the potential to affect the growth of specific oral bacteria. Polymorphisms in genes such as leucine zipper motif isoform 2 (APPL2) and glucose transporter 9 (SLC2A9), which are closely linked to obesity and insulin resistance, have been associated with changes in the abundance of various oral bacteria, including Prevotella species. These genetic variations may regulate bacterial growth through mechanisms involving host microRNAs [123,124]. On the other hand, alterations in microbiota composition and function have been also implicated in disease pathogenesis through epigenetic modifications, such as DNA methylation, histone changes, and regulation by noncoding RNAs [125].”

Comments 2:     Similarly, external confounders that play a role in the link between oral microbiome and CVD where not mentioned. For example, smoking and social determinants of health.

Response 2: Thanks again for this valuable comment. As suggested, we have now incorporated one additional paragraph to discuss how these external confounders, especially smoking and social determinants, impact the oral microbiome and CVD risk [Line 499- 507].

Comments 3:     Please add a reference to line 211 to the end of the sentence: “Prevotella intermedia produces lipase during bacterial invasion…”

Response 3: Thanks. We have added the reference.

Comments 4:     There is some additional recent literature that could be considered about metabolomics, microbiome, oral disease, and cardiovascular disease intersection:

https://www.ahajournals.org/doi/full/10.1161/JAHA.123.033350

https://link.springer.com/article/10.1007/s00784-021-04083-4

https://journals.sagepub.com/doi/10.1177/23800844241280383?url_ver=Z39.88-2003&rfr_id=ori:rid:crossref.org&rfr_dat=cr_pub%20%200pubmed

Response 4: Thanks. We have incorporated the contents of these recent studies in this revision.  (Line 69, reference [15]  ; Line 169, reference [44] ; and Line 109, reference [30] )

Comments 5:      Please add a footnote to Table 1 to describe abbreviation of "N" in specific quality features.

 Response 5: Thanks. We have added the footnote in Table 1.

Reviewer 2 Report

Comments and Suggestions for Authors

I thank the authors and the editor for giving me the opportunity to read and review this paper. The present narrative review evaluates the role of the oral microbiome in the pathogenesis of CVD. The authors claim that although the relationship between the gut microbiome and CVD has attracted considerable research attention the role of the oral microbiome remains less understood. Previously reports associations between the oral microbiome and CVD, has been focused of the known periopathogens Porphyromonas gingivalis (Pg), Fusobacterium nucleatum (Fn), and Aggregatibacter actinomycetemcomitans (Aa). Microbial metabolites, such as SCFAs, TMAO, H2S, NO and LPS have also been addressed in relation to CVD or the gut microbiome., The authors conclude that fundamental knowledge will support the development of targeted interventions and effective therapies to prevent or reduce the risk to develop clinical CVD events.

General aspects:

The organization of this narrative review is fine, however, it still lacks several pieces of information and a lot of editorials mistakes are still in the text. For more detailed aspects, see specific points below.

Specific points:

Line 27             “Specific toxins”, not only LPS, suggest that the well-studied LtxA from Aa should be mentioned here?

Line 63             Pro-inflammatory cytokine are not of microbial origin. Suggest that the authors extent the text so the reader can understand what is of bacterial origin and from the host-response.

Line 93-96      What is known for the relation with periodontitis and bacteria or bacterial components (vesicles, lps, SCFA etc) presence I the systemic circulation?

Line 117          Aggregatibacter taxa can also grow in presence of oxygen, and therefore an early colonizer of the periodontium. The other three bacterial taxa is strict anaerobic and mainly associated with deep periodontal pockets.

Line 118-123  Here we need references, and additional information of which microbial species that have been detected. In addition, was it DNA-based methods or cultivation of viable bacteria.

Line 138          Again, all bacteria associated with systemic presence is not anaerobic, taxa from Streptococci and Aggregatibacter is commonly detected in CVD cases.

Line 142          Use P. gingivalis after the bacterial species name has been written for the first time. Check all full species name throughout the whole manuscript for this aspect.

Line 192-194  Here we need a reference for the cytotoxicity, as well as the cellular mechanisms induced.

Line 291-318 Define is the authors believe that oral bacteria can be translocated systemically trough gut epithelial barrier?

Line 300          Use the current name for Aggregatibacter actinomycetemcomitans. Check also the reference (67), which not is relevant here.

Line 356-358 Is the levels of pro-inflammatory cytokines systemically a leakage from the periodontium or an effect of systemic bacterial translocation. Suggest the authors add a hypothesis.

Line 456          Write H2S correct!

Missing information:

Suggest that the authors better organize the structure of the manuscript in the aspect microbial detection. Define when samples from saliva, subgingival plaque, blood or various tissues. In addition, is bacterial DNA detected or have cultivation techniques been employed.

Moreover, can the authors recommend periodontal treatment to prevent CVD risk? Is reported from intervention studies that remission of periodontitis decrease CVD risk? Suggest that the authors should add information and speculations about this aspect.

                           Finally, many population studies have used humoral immunoreactivity to confirm individual reaction to a specific bacterium. Suggest that the authors should include a section of humoral immunoreactivity against selected oral bacteria.

Author Response

Thank you very much for taking the time to review this manuscript. We greatly appreciate these constructive suggestions. We have revised the manuscript accordingly, added the additional information as recommended, and improved the editorial aspects throughout the text. Please find the detailed responses below, with corresponding revisions highlighted in red font in the re-submitted files.

Comments 1:      Line 27. “Specific toxins”, not only LPS, suggest that the well-studied LtxA from Aa should be mentioned here?

Response 1: We appreciate this constructive comment. As suggested, in this revision, we have included additional information related to LtxA. We have also provided a more detailed description of LtxA, covering its origin  source (i.e., Aggregatibacter actinomycetemcomitans), the established mechanisms by which it acts, and the recognized effects it exerts on the host, on Lines 208– 213.

Comments 2:   Line 63. Pro-inflammatory cytokine are not of microbial origin. Suggest that the authors extent the text so the reader can understand what is of bacterial origin and from the host-response.

Response 2: Thanks. We totally agree that clarifying the host origin of these pro-inflammatory cytokines would be helpful. As suggested, we have revised the text to enhance clarity for the reader, as indicated in Line 61- 66.

Comments 3:  Line 93-96. What is known for the relation with periodontitis and bacteria or bacterial components (vesicles, lps, SCFA etc) presence I the systemic circulation?

Response 3: Thanks. The relationships between periodontitis and the presence of bacterial components (vesicles, LPS, SCFAs, etc.) in the systemic circulation remain not fully elucidated. Here we have outlined the most well-established evidence currently available on this topic; however, further studies are essential to clarify these complex interactions. Current evidence suggests that periodontal infections may facilitate the translocation of bacteria and their components into the bloodstream due to increased vascular permeability at infected sites, particularly in cases of periodontitis and other oral inflammatory conditions. In Line 336 (section 4.2), we discuss the mechanisms through which these bacteria and bacterial elements may influence systemic inflammation and vascular health. Furthermore, LPS, recognized as a potent immunostimulatory molecule, can translocate from the oral cavity into the circulation, contributing to endotoxemia and endothelial dysfunction. Additional details on the role of LPS are provided in Line 353- 369. Altogether, further research is warranted to fully elucidate the complex interactions mentioned above and their broader implications.

Comments 4:  Line 117. Aggregatibacter taxa can also grow in presence of oxygen, and therefore an early colonizer of the periodontium. The other three bacterial taxa is strict anaerobic and mainly associated with deep periodontal pockets.

Response 4: Many thanks for this constructive suggestion. We have revised the manuscript accordingly, adding more descriptions that emphasize the “facultative anaerobic” feature of Aggregatibacter [Line 125]. Additionally, we have also included more descriptions in the sections introducing each specific bacterium (sections 2.2.1-2.2.5).

Comments 5:  Line 118-123. Here we need references, and additional information of which microbial species that have been detected. In addition, was it DNA-based methods or cultivation of viable bacteria.

Response 5: Thank you. We agree that providing details on microbial species and detection methods is beneficial. Therefore, in Table 1, we have listed the 18 most relevant studies (with all references included) and incorporated information on microbial species (column “Findings” in Table1) and the specific methods used (column “Methods” in Table1). Furthermore, for the most extensively studied five bacterial taxa, we have dedicated individual sections to each (see sections 2.2.1 - 2.2.5) to provide a comprehensive overview.

Comments 6:  Line 138. Again, all bacteria associated with systemic presence is not anaerobic, taxa from Streptococci and Aggregatibacter is commonly detected in CVD cases.

Response 6: Thanks again. We have revised the relevant sentences to make it more clear (Line 143). Additionally, we have also included more descriptions in the sections introducing each specific bacterium (sections 2.2.1-2.2.5). For example, we highlighted the “facultative anaerobic” feature of Aggregatibacter in section 2.2.3.

Comments 7:  Line 142. Use P. gingivalis after the bacterial species name has been written for the first time. Check all full species name throughout the whole manuscript for this aspect.

Response 7: Thanks. As suggested, we have made the appropriate changes throughout the whole text.

Comments 8:  Line 192-194. Here we need a reference for the cytotoxicity, as well as the cellular mechanisms induced.

Response 8: Thanks. We have added a more detailed description with references for the cytotoxicity and the associated cellular mechanisms. [Line 208-216]

Comments 9:  Line 291-318 Define is the authors believe that oral bacteria can be translocated systemically trough gut epithelial barrier?

Response 9: Thanks for this insightful comment. Based on current literature, we believe it is plausible that bacteria, including some of oral origin bacteria, may translocate systemically via the gut epithelial barrier. In addition to the direct entry of oral bacteria into the bloodstream through periodontal vasculature, studies suggest that interactions between oral and gut microbiota can affect gut permeability. Dysbiosis in the oral microbiome has been associated with changes in gut health, potentially compromising gut barrier integrity (more details in section 3). This compromised barrier may allow bacterial translocation, including oral bacteria or their byproducts, through the gut epithelium into systemic circulation. Nevertheless, research on oral-gut microbiota interactions and bacterial translocation through the gut epithelial barrier remains in early stages, and further studies are needed to provide more substantial evidence. As suggested, we have elaborated on this hypothesis in the revised manuscript [Line 329-334).

Comments 10:  Line 300. Use the current name for Aggregatibacter actinomycetemcomitans. Check also the reference (67), which not is relevant here.

Response 10: Thanks. As suggested, in this revision, we have standardized the use of the current name Aggregatibacter actinomycetemcomitans throughout the manuscript, ensuring that the old name no longer appears. We have also updated the references to ensure their relevance.

Comments 11:  Line 356-358. Is the levels of pro-inflammatory cytokines systemically a leakage from the periodontium or an effect of systemic bacterial translocation. Suggest the authors add a hypothesis.

Response 11: Thanks for this insightful suggestion. Indeed, this is an interesting and complex area with limited existing literature. Most studies on this topic to date are observational (e.g., cross-sectional epidemiological studies with small sample sizes). Further research, particularly mechanistic studies, is necessary to deepen our understanding in this field. Additionally, many factors could potentially influence pro-inflammatory cytokine levels, and it is likely that the result may from a combination of multiple factors, including periodontium leakage and potential bacterial translocation, etc. Integrative analysis using multi-omics techniques (e.g., metagenomics and proteomics) could enhance our understanding in this area. As suggested, we have incorporated this point in the revised manuscript [Line 402-406].

Comments 12:  Line 456. Write H2S correct!

Response 12: Thanks. We have corrected the format throughout the text.

Comments 13:  Suggest that the authors better organize the structure of the manuscript in the aspect microbial detection. Define when samples from saliva, subgingival plaque, blood or various tissues. In addition, is bacterial DNA detected or have cultivation techniques been employed.

Response 13:  We thank the reviewer for this constructive suggestion. We have organized this information into Table 1, where we have a dedicated column, "Sample Site," specifying the sample sources for each study, such as saliva, buccal mucosa, tongue, gingiva, coronary artery tissue, etc. Additionally, a "Method" column in Table 1 highlights the specific microbial detection techniques used, such as 16S rRNA sequencing (with sequencing region specified, e.g., V3-V4), targeted PCR sequencing, and DNA hybridization, etc. Since the majority of current studies rely on bacterial DNA related techniques, we have also explicitly noted in the main text those studies that employed traditional cultivation techniques.

Comments 14:  Moreover, can the authors recommend periodontal treatment to prevent CVD risk? Is reported from intervention studies that remission of periodontitis decrease CVD risk? Suggest that the authors should add information and speculations about this aspect.

Response 14: Thanks. To the best of our knowledge, there are very few studies on this topic, and those available did not specifically focus on the oral microbiome's role. According to the literature (Hopkins et al., 2024), to date, no well-powered studies of the effects of periodontal treatment on hard cardiovascular disease end points (myocardial infarction, stroke, cardiovascular death) have been conducted.  The closest available reports focus on general oral hygiene  (rather than directly on periodontal interventions). Park et al. reported that brushing teeth at least once per day or undergoing professional dental cleaning once per year was associated with a 9% and 14% reduction in cardiovascular risk, respectively (Park et al.2019), suggesting that adherence to oral hygiene may help lower cardiovascular disease risk.
Further studies are warranted to determine whether periodontal treatment holds the potential to reduce CVD risk and to reveal how mechanisms related to the oral microbiota contribute to this possible protective effect. We included this content in section 2.1 [Line 116].

Reference:
[1] Hopkins, S.; Gajagowni, S.; Qadeer, Y.; Wang, Z.; Virani, S.S.; Meurman, J.H.; Krittanawong, C. Oral Health and Cardiovascular Disease. The American Journal of Medicine 2024, 137, 304-307.

[2] Park, S.; Kim, S.; Kang, S.; Yoon, C.; Lee, H.; Yun, P.; Youn, T.; Chae, I. Improved Oral Hygiene Care Attenuates the Cardiovascular Risk of Oral Health Disease: A Population-Based Study from Korea. Eur Heart J 2019, 40, 1138-1145.

Comments 15:  Finally, many population studies have used humoral immunoreactivity to confirm individual reaction to a specific bacterium. Suggest that the authors should include a section of humoral immunoreactivity against selected oral bacteria.

Response 15: Thanks. In this revision, we have incorporated relevant contents, summarizing literature that examines humoral immunoreactivity to confirm individual responses to specific bacteria. In particular, we have included discussions on key bacteria, such as Porphyromonas gingivalis [Line 164-] “Studies based on serum concentrations of anti-Porphyromonas antibodies have shown that several Porphyromonas species are positively associated with cardiovascular risk [44,45].”;  Fusobacterium nucleatum [Line 181-] “Recent research indicated that antibodies to F. nucleatum may be linked to cardiovascular events. A study found that higher antibody level against F. nucleatum was significantly associated with higher incidence of coronary heart disease, after adjustment for established risk factors [53].”; and Aggregatibacter actinomycetemcomitans [Line 205-] “One study found that Elevated serum concentrations of anti-A. actinomycetemcomitans antibodies were linked to a greater cardiovascular risk and future coronary heart disease events [45].”.

Round 2

Reviewer 2 Report

Comments and Suggestions for Authors

The revised version is ubstantially improved and the review response fine. However, check mor minor edit mistakes, like line 205 with capital E in elevated.